# Atrial Fibrillation: A New Indicator for Advanced Colorectal Neoplasia in Screening Colonoscopy

**DOI:** 10.3390/jcm8071083

**Published:** 2019-07-23

**Authors:** Peter C. Kahr, Sabrina Hammerl, Ursula Huber-Schönauer, Christian M Schmied, Laurent M. Haegeli, Slayman Obeid, Sarah Eder, Sebastian Bachmayer, Elmar Aigner, Christian Datz, David Niederseer

**Affiliations:** 1Department of Cardiology, University Heart Center, 8091 Zurich, Switzerland; 2Department of Internal Medicine, University Hospital, 8091 Zurich, Switzerland; 3Department of Internal Medicine, Teaching Hospital of the Paracelsus Medical University Salzburg, General Hospital Oberndorf, Paracelsus Medical University Salzburg, Oberndorf 5110, Austria; 4Division of Cardiology, Medical University Department, Kantonsspital Aarau, 5000 Aarau, Switzerland; 5First Department of Internal Medicine, Paracelsus Medical University Salzburg, Salzburg 5020, Austria

**Keywords:** atrial fibrillation, colorectal cancer, screening, prevention

## Abstract

Background: Colorectal cancer (CRC) is a common and potentially preventable malignancy. Evidence has emerged that coronary artery disease patients are at increased risk for developing CRC by shared risk factors. Here we investigated an association between CRC and atrial fibrillation (AF), a surrogate marker of cardiovascular risk, in the setting of routine screening colonoscopy. Methods: We investigated 1949 asymptomatic participants (median age 61 [54–67] years, 49% females) undergoing screening colonoscopy within the SAKKOPI registry (Salzburg Colon Cancer Prevention Initiative). Forty-six participants with AF (2.4%) were identified, and colonoscopy findings were compared to non-AF participants. Propensity Score Matching (PSM) was used to create 1:1 and 3:1 age- and gender-matched couples. Results: Abnormal findings on screening colonoscopy (any form of adenoma or carcinoma) were more common in AF participants with an odds ratios (OR) of 2.4 [1.3–4.3] in the unmatched analysis, and 2.6 [1.1–6.3] and 2.0 [1.1–4.0] in the 1:1 and 3:1 matched groups, respectively. Correspondingly, the odds of finding advanced adenomas or carcinomas was elevated about three-fold across the different matched and unmatched analyses (OR 3.3 [1.1–10.8] for 3:1 matched participants). At the same time, the prevalence and number of colonic lesions were significantly higher in AF participants (63.0% vs. 33.4% for 3:1 matched participants, *p* < 0.001). Non-CRC related findings on colonoscopy, like diverticulosis, were non-different between groups. Conclusion: Participants with AF had a higher burden of advanced premalignant adenomas and CRC in routine colonoscopy screening. Our data suggest that practitioners should monitor the CRC screening status, especially in AF patients.

## 1. Introduction

Colorectal cancer (CRC) has a high burden of disease. It accounts for about 12% of all cancer-related deaths in Europe, corresponding to 3% of all-cause mortality or about 150,000 annual deaths (source: Eurostat, data for 2015, last update 17 October 2018) [1]. CRC continues to rank as the second (male) and third (female) most common cause of death from cancer across developed countries. While male sex and increasing age are common risk factors for CRC worldwide, its prevalence is highest in Western countries and historically low in Africa and South-Central Asia. This reflects the contribution of risk factors, including the consumption of red/processed meat and alcohol, overweight and physical inactivity, and smoking. While CRC incidence is increasing in some countries due to the adoption of a more Western lifestyle associated with the above-mentioned risk factors (e.g., Israel and Eastern Europe [2]), mortality is decreasing in other countries due to the reduction of risk factors, improved treatment modalities, and CRC screening allowing the removal of premalignant adenomas [3,4]. It has been calculated that, among these three factors, CRC screening has had the largest effect, explaining 53% of the mortality reduction between 1976 and 2000 [4]. While different approaches are utilized for CRC screening, including an immunochemical and a guaiac-based fecal occult blood test (FOBT), fecal DNA tests, computed tomography colonoscopy, and double-contrast barium enema, colonoscopy is accepted as the gold standard screening method in the United States and Europe [5,6].

Similar to CRC, atrial fibrillation (AF) is a widespread disease in the Western world, being the most common heart rhythm disorder in adults affecting about 2–3% of the general population in Europe [7] and more than 33 million patients worldwide [8]. Although there is an inherited component to the development of atrial fibrillation, particularly in young patients with “lone AF”, AF often occurs secondary to structural changes of the atria that are caused by other forms of cardiovascular disease. Hence, cardiovascular risk factors such as obesity, arterial hypertension, diabetes mellitus, and smoking are also associated with the development of AF [9].

Recently, it has been acknowledged that CRC and cardiovascular diseases, particularly coronary artery disease, share several risk factors [10]. We previously demonstrated that cardiovascular risk factors and the presence of coronary artery disease correlate with the detection of colorectal neoplasms on screening colonoscopy [11]. In the current study, we aimed to investigate whether participants with AF have an increased risk of colorectal neoplasms on screening colonoscopy.

## 2. Methods

### 2.1. Subjects

Participants free of gastrointestinal symptoms were recruited through the screening program at the Department of Internal Medicine, Oberndorf (Teaching Hospital of the Paracelsus Medical University Salzburg, Austria) between 2010 and 2014 [11]. Ethical approval for the SAKKOPI registry (Salzburg Colon Cancer Prevention Initiative) was given by the local ethics committee (Ethikkommission des Landes Salzburg, approval no. 415-E/1262/2-2010). All consecutive patients aged 45–79 years were included, with no exclusion criteria. The screening was performed as indicated by national screening recommendations [12], and informed consent was obtained from all participants.

### 2.2. Patient Assessment

Study participants were examined twice: blood collection after overnight fasting and physical examination on day 1, screening colonoscopy on day 2. A detailed medical history was obtained from every participant, including the current medical regimen and the presence of cardiovascular comorbidities. Cardiovascular risk was assessed by calculating the Heart Score of the European Society of Cardiology (ESC) [13] based on current cholesterol levels and blood pressure as well as demographic variables. The ESC Heart Score estimates the 10-year risk of cardiovascular death.

Participants with AF were identified based on their current reported medication (Figure 1). If participants where on oral anticoagulation (OAC) or had an increased international normalized ratio (INR), further information was gathered from healthcare providers regarding the underlying condition, including review of their ECG. This way, AF participants were separated from those on OAC for pulmonary embolism or deep vein thrombosis (DVT). In addition, INR was measured in each participant to identify participants who possibly fail to report OAC, but actually, take OAC for AF or other conditions. Since non-vitamin K oral anticoagulants were only scarcely available in Austria at the time that the cohort was screened, most patients were taking Acenocoumarol (Sintrom^R^) or Phenprocoumon (Marcoumar^R^).

Colonoscopic findings were classified as tubular adenoma, advanced adenoma (i.e., villous or tubulovillous features, size ≥ 1 cm or high-grade dysplasia, AN) or carcinoma (CRC; colorectal cancer) after a combined analysis of macroscopic and histological results [14,15]. Only the highest-scoring lesion was counted to allocate patients. We defined the presence of any form of adenoma up to carcinoma as an abnormal finding. Hyperplastic polyps per se were not classified as abnormal but were investigated separately. Consequently, a normal colonoscopy was defined as the absence of adenoma and carcinoma.

### 2.3. Propensity Score Matching and Statistical Analysis

We used SPSS version 25 (IBM) for data management and statistical calculations. For comparison of categorical variables, we used a contingency χ2 test. Linear variables were compared between groups using the nonparametric Mann–Whitney U-test. A *p* < 0.05 was accepted as statistically significant, and values were given as median [interquartile range] or n (percentage). We estimated odds ratios (ORs) with 95% confidence intervals (CI) by univariate logistic regression analysis. Haldane–Anscombe correction was used for calculation of odds ratios in small frequency samples when an event did not occur in one of the groups.

We used propensity score matching (PSM) to create a non-AF participant group that resembles the demographic characteristics of the AF subpopulation since significant differences in demographic parameters between AF participants and the remaining population were identified. We used logistic regression analysis in the entire cohort to calculate a predicted probability (propensity score) and controlled for age and sex. Thereafter, AF participants were randomly matched in a 1:1 and 3:1 ratio to non-AF participants with equal propensity scores (acceptable differences <0.01).

## 3. Results

### 3.1. Study Population and Cardiovascular Risk

Our screening population included a total number of 1949 participants (median age 61 years, 49% female, see Table 1 for demographic details). Of these, 73 participants reported being on OAC, and 60 participants were found to have INR-elevations >1.2 (Figure 1). Based on the in-detail review of the participant histories, including ECG (electrocardiogram), 46 participants (2.4%) were found to have a history of AF. Among these, there were no cases of valvular AF (i.e., moderate to severe mitral stenosis or mitral valve prosthesis), and their age ranged between 57 and 79 years (median 72 years). The remaining participants on OAC were treated either for deep vein thrombosis (DVT) or pulmonary embolism, or had spontaneous INR elevations. The demographics for participants in the AF group were significantly different from the general population in several aspects: AF participants were predominantly male and older (Table 1). Concordantly, clinical cardiovascular risk factors (e.g., diabetes mellitus and arterial hypertension) were more common, and risk scores were significantly higher in AF participants compared to the rest of the unmatched screening population. Hence, propensity scoring was performed to match participants based on age and sex in a 1:1 and 3:1 ratio (corresponding to 46 and 138 matched non-AF participants, respectively). After matching, the demographic parameters were found to be non-different between AF and non-AF participants (Table 1).

Compared to the unmatched population, AF participants had significantly higher cardiovascular risk as assessed by the ESC Heart score. Also, manifested cardiovascular disease and comorbidities were significantly more common (coronary artery disease, arterial hypertension, diabetes) in AF participants. After PSM, diabetes and coronary artery disease continued to be more prevalent in the AF group. Assessment of overall cardiovascular risk by the ESC Heart Score; however, demonstrated an increased 10-year risk estimate in the matched non-AF groups compared to AF participants.

### 3.2. CRC Screening Results

Overall, abnormal findings on screening colonoscopy were found in 28.4% of participants. Compared to non-AF participants, this prevalence was significantly higher in participants with AF (47.8%, Table 2). The odds ratio for abnormal screening colonoscopy in the AF group was 2.4 [1.3–4.3] compared to all non-AF participants which were 2.6 [1.1–6.3] compared to the 1:1 matched group and 2.0 [1.1–4.0] compared to the 3:1 matched group (Figure 2). This difference was mainly driven by the significantly increased finding of carcinomas with/without advanced adenomas. The odds of finding carcinoma in AF participants was 18.0 times higher when compared to all non-AF participants and 9.8/6.5 times higher when compared to a 1:1/3:1 PSM-matched group of non-AF participants. 

While there was a significant difference in the number of tubular and borderline significant difference in the finding of advanced adenomas between the AF and unmatched non-AF groups, these differences were abolished by matching (Appendix A). On the other side, we found a significantly increased prevalence of colonic lesions upon screening colonoscopy in AF participants: while 67.4–73.9% of non-AF participants were free of any colonic lesion, depending on the type of matching, only 37.0% of AF participants were free of colonic lesions on colonoscopy (Appendix A). Diverticulosis was more common, of higher severity, and extending further in the colon in AF-participants compared to the unmatched group of non-AF participants (Appendix A). However, after matching, we did not observe any differences in the prevalence or severity of diverticulosis between AF and non-AF groups (Appendix A).

## 4. Discussion

In this study, we report a higher prevalence of advanced colorectal adenomas and carcinomas in patients with atrial fibrillation undergoing screening colonoscopy.

### 4.1. Study Population and Matching

To the best of our knowledge, this study is the first to describe an increased prevalence of abnormal findings on screening colonoscopy in AF participants. The reported prevalence of AF in this study (2.4%) corresponds well with previously published numbers for the same age group in Germany (2.5%) [16]. AF participants had significantly different demographics from the rest of the screening population in this study (male gender and older age); therefore, we used propensity score matching (PSM) to create a comparable subgroup of non-AF participants. While we did not hypothesize a direct (causal) relationship between AF and CRC, it was our intention to investigate whether AF would be associated with abnormal findings on CRC screening, because it is a surrogate parameter for global cardiovascular risk. Hence, we did not include other markers of cardiovascular risk, including established risk scores (e.g., ESC Heart Score) or other forms of manifested cardiovascular disease (e.g., coronary artery disease) in the propensity score. After matching, AF participants were found to have a lower risk of cardiovascular mortality as estimated by the ESC Heart Score in comparison to the matched non-AF groups. Hence, we conclude that matching based on age and gender was only sufficient to correct for major risk factors of heart disease in this cohort. Furthermore, the prevalence and severity of diverticular disease of the colon, an incidental finding on screening colonoscopy, was found to be equivalent between groups (Appendix A). This finding highlights the good overall matching of participants in our study. While the presence of diverticulosis is also strongly age- and gender-dependent, it is also influenced by cardiovascular risk factors that cumulated by a Western way of living [17].

### 4.2. Obstacles to CRC Screening in AF Patients

While it is widely accepted that the implementation of a CRC screening program reduces the burden of CRC, there is still controversy regarding the ideal way of screening (FOBT vs. colonoscopy). In AF patients, both screening methods are faced with additional concern—while colonoscopy is a routine procedure with a minimal rate of adverse events, especially in the setting of an elective screening procedure, the risk of gastrointestinal events (bleeding and perforation) is significantly increased in AF patients [18]. This can be partially explained by most AF patients being on oral anticoagulation agents. Although clinical practice guidelines have been published [19,20], physicians are still concerned about the ideal peri-procedural anticoagulation in the individual patient that minimizes both the risk of peri-procedural stroke and bleeding complications. It has been shown that, especially non-cardiologists, perform excessive bridging with low molecular weight heparin [21], particularly in patients at low cardioembolic risk, which is associated with an increased risk of bleeding [22]. On the other side, anticoagulation may interfere with CRC screening by FOBT, resulting in increased [23,24,25,26] or reduced [25] positive predictive value of the screening. In our study, none of the screened patients underwent FOBT screening prior to endoscopy, and symptomatic subjects with prior rectal bleeding were excluded so that a strictly asymptomatic cohort was investigated in this study.

### 4.3. Causality

It is relatively easy to name plausible reasons why cancer patients have an increased risk of developing AF or being diagnosed with AF, especially during the initial period after diagnosis; surgical, as well as medical treatment may trigger new-onset AF as patients are under extensive medical surveillance/monitoring during this period of their life [27,28]. It is more difficult to explain the reverse relationship: why would AF patients have a higher rate of cancer? This finding of our study is in line with previously published data from Denmark [29], from the Women’s Health Study [30] and the Explorys platform [31], which have shown an increased rate of overall cancer diagnoses in new-onset AF patients. Possible explanations include increased detection due to above-average medical surveillance or bleeding from cancer if under anticoagulation, as well as unknown systemic or genetic factors and increased susceptibility to symptoms by the individual patient [32]. However, shared risk factors are most likely to explain this association.

### 4.4. AF—an Indicator of Lifestyle Associated Disease

While lone or familial AF in young and otherwise healthy patients is a rarity, several risk factors for AF have been identified. In addition to the “classical” risk factors, higher age and male sex, evidence has emerged that the development of AF is strongly associated with lifestyle factors. Current and former smoking increases the risk of developing AF by a factor of 1.5 [33], while a 1-unit increase of BMI (body mass index) elevates the risk to have AF by 4% [34]. On the other side, weight loss has been demonstrated to reduce long-term freedom from AF (A Long-Term Follow-Up Study: the LEGACY study) [35] and the progression of AF (PREVEntion and regReSsive Effect of weight-loss and risk factor modification on Atrial Fibrillation: the REVERSE-AF study [36]. In addition, cardiorespiratory fitness training has been demonstrated to further reduce the burden and symptom severity in overweight AF patients beyond the effect of weight loss (Impact of CARDIOrespiratory FITness on Arrhythmia Recurrence in Obese Individuals With Atrial Fibrillation: The CARDIO-FIT Study) [37]. Similar to arterial hypertension, AF can be seen as an indicator of lifestyle-associated cardiovascular risk. Although the rates of smoking and arterial hypertension were not different between AF patients and matched non-AF participants in our cohort, a higher presence of coronary artery disease and diabetes mellitus (and AF itself) demonstrated a true difference in cardiovascular risk between both groups.

At the same time, the risk of developing CRC is highly modified by lifestyle: smoking and obesity increase the risk of developing CRC by approximately 20%, while high physical activity is protective according to a meta-analysis of >25,000 CRC patients [38]. Although this study is not able to prove the hypothesis of a causal relationship between lifestyle, atrial fibrillation, and colorectal cancer risk, our data highlight the importance of a generalist and interdisciplinary approach towards AF as a surrogate parameter for general health risk factors, contributing to cardiovascular disease, as well as colorectal cancer. Although there may be a hesitancy to perform a colonoscopy in AF patients under oral anticoagulation, our observations demonstrate the importance of CRC screening, particularly in this patient population at increased risk for malignancies. While we would not recommend screening AF patients outside of the established screening guidelines, general practitioners and cardiologists should monitor this population at risk for adequate CRC screening.

## 5. Limitations

Given the relatively low prevalence of AF in the general population, the present study included only a relatively small number of AF participants despite screening almost 2000 individuals by colonoscopy. Further information from larger datasets is certainly warranted to confirm the findings of this study. Since we did not ask for AF specifically in the questionnaire that was handed out prior to colonoscopy, our identification is based on a retrospective analysis of participants taking OAC and having abnormal INR values. Hence, we may have missed individuals that have failed to report being on OAC with normal INR values and those with AF and low stroke risk scores not under anticoagulant treatment. Due to the presence of asymptomatic, incidental AF, however, the latter limitation also applies to prospective studies comparing AF and non-AF populations. 

Our study includes only individuals undergoing a screening for colonoscopy. Hence, symptomatic patients (with anemia, melena, stool abnormalities) are not included in the population, and we can only speculate about a potential association between AF and CRC in these patients. While we assume an association between AF, CRC and lifestyle parameters as a common source, our analysis did not include lifestyle parameters beyond weight/BMI and smoking history.

## Figures and Tables

**Figure 1 jcm-08-01083-f001:**
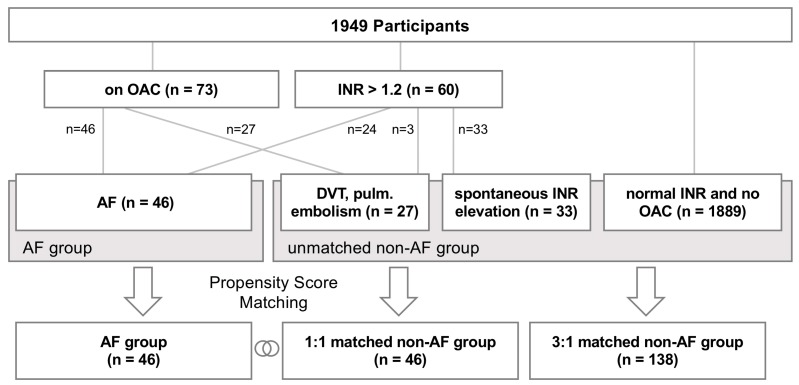
Patient Inclusion. In this retrospective study, 1949 participants in a CRC (colorectal cancer) screening program were included. Participants with AF (atrial fibrillation) were identified by investigation of detailed medical records those individuals under treatment with OAC (oral anticoagulation) and with elevated INR (international normalized ratio). This resulted in a cohort of 46 patients with AF, which were matched with 46 (1:1) and 138 (3:1) non-AF participants utilizing age- and gender-based propensity scores.

**Figure 2 jcm-08-01083-f002:**
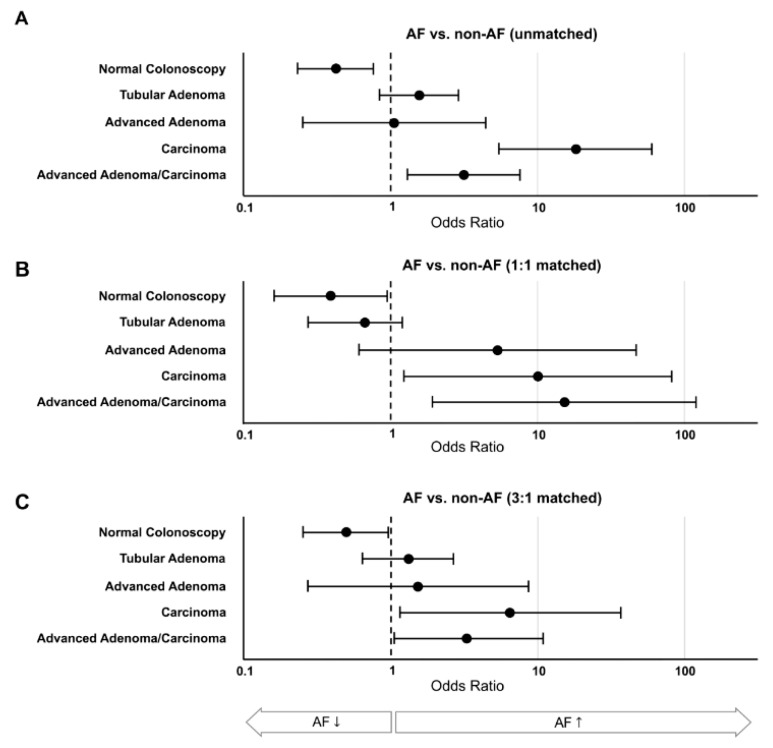
CRC (colorectal cancer) Screening Results. Comparing AF (atrial fibrillation) and non-AF participants (**A** unmatched, **B** 1:1 matched and **C** 3:1 matched), odds ratios for a normal colonoscopy, tubular adenoma (benign), advanced adenoma (premalignant), and carcinoma were calculated. The odds ratio for a combined endpoint of advanced adenoma and carcinoma was calculated because of its clinical significance. While normal colonoscopy was more significantly common in non-AF controls, there was a significantly increased risk of advanced neoplasias in AF participants across all comparisons.

**Table 1 jcm-08-01083-t001:** Demographics and Cardiovascular Risk Factors.

	All Participants	Non-AF Group			AF Group	Statistical Comparison AF vs. Non-AF		
		Unmatched	1:1 Matched	3:1 Matched		Unmatched	1:1 Matched	3:1 Matched
***PARTICIPANTS***	**1949**	**1903**	**46**	**138**	**46**			
Female	957 (49.1%)	948 (49.8%)	9 (19.6%)	28 (20.3%)	9 (19.6%)	***p* < 0.001**	*p* = 1	*p* = 0.915

Age (years)	61 [54–67]	60 [5–67]	72 [68–75]	72 [68–75]	72 [68–75]	***p* < 0.001**	*p* = 1	*p* = 0.955
Range	46–79	46–79	57–79	57–79	57–79			
BMI (kg/m2)	26.8 [24.1–29.8]	26.7 [24.1–29.8]	26.4 [24.3–29.4]	27.1 [24.9–30.3]	27.8 [24.0–30.4]	*p* = 0.264	*p* = 0.276	*p* = 0.984
***COMORBIDITIES***								
ESC SCORE	0.02 [0.01–0.05]	0.02 [0.01–0.05]	0.07 [0.05–0.11]	0.07 [0.04–0.10]	0.05 [0.03–0.09]	***p* < 0.001**	***p* = 0.039**	***p* = 0.047**
Coronary Artery Disease	112 (6%)	99 (5%)	5 (11%)	20 (14.5%)	13 (28%)	***p* < 0.001**	***p* = 0.036**	***p* = 0.035**
Diabetes Mellitus	303 (16%)	282 (15%)	9 (20%)	26 (21.8%)	21 (46%)	***p* < 0.001**	***p* = 0.008**	***p* < 0.001**
Arterial Hypertension	1237 (64%)	1224 (65%)	34 (74%)	104 (75.4%)	29 (63%)	*p* = 0.864	*p* = 0.262	*p* = 0.106
Smoking Ever Current	545 (48%) 192 (17%)	533 (48%) 189 (17%)	22 (48%) 5 (11%)	55 (39.9%) 14 (10.1%)	14 (31%) 6 (13%)	*p* = 0.085 *p* = 0.333	*p* = 0.087 *p* = 0.748	*p* = 0.253 *p* = 0.584

Comparison of baseline participant characteristics for the AF (atrial fibrillation) and the non-AF groups, including an unmatched as well as 1:1 and 3:1 propensity score matched groups. The numbers are given as a number (percentage) or median [interquartile range]. The level of significance was set to 0.05 and significant differences were marked bold.

**Table 2 jcm-08-01083-t002:** Colonoscopy Results.

	All Participants	Non-AF Group			AF Group	Statistical Comparison AF vs. Non-AF		
		Unmatched	1:1 Matched	3:1 Matched		Unmatched	1:1 Matched	3:1 Matched
**Number of Participants**	**1949**	**1903**	**46**	**138**	**46**	**-**	**-**	**-**
***CRC Screening Results***								
Normal Colonoscopy	1395 (71.6%)	1371 (72.1%)	34 (73.9%)	95 (68.8%)	24 (52.2%)	**0.003**	**0.031**	**0.041**
Tubular Adenoma	503 (25.8%)	487 (25.6%)	12 (26.1%)	40 (29.0%)	16 (34.8%)	0.160	0.365	0.459
Advanced Adenoma	81 (4.2%)	79 (4.2%)	0	4 (2.9%)	2 (4.3%)	0.948	0.153	0.632
Carcinoma	14 (0.7%)	10 (0.5%)	0	2 (1:4%)	4 (8.7%)	**<0.001**	**0.041**	**0.017**
Adv. Adenoma & Carcinoma combined	93 (4.8%)	87 (4.6%)	0	6 (4.3%)	6 (13.3%)	**0.008**	**0.013**	**0.039**

The numbers are given as a number (percentage). The level of significance was set to 0.05 and significant differences are marked in bold.

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
