# Peer review of "Atrial Fibrillation: A New Indicator for Advanced Colorectal Neoplasia in Screening Colonoscopy"

_jcm, 2019, doi:10.3390/jcm8071083_

Round 1
Reviewer 1 Report
The purpose of this study by Kahr et al to investigate whether there is an association between CRC and atrial fibrillation in the setting of routine colonoscopic screening. The authors to be congratulated on this work and presenting new clinical encounters for aFIB population. Screening colonoscopy with any form of adenoma or carcinoma were more common in AF participants with odds ratios (OR) being >2 in both unmatched and matched group analysis with no different in Non-CRC related findings on colonoscopy like diverticulosis. In their conclusion afib population found to have higher predisposition to have advanced premalignant adenomas and CRC in routine colonoscopy screening. The implications of this findings is real world practice can be quite interesting where a fib patient can be referred to colonoscopy and vice versa. The paper is well written, and I do not have further comments but:
- Can the author disclose that this work has been presented in the ESC 2018! Understand this is the manscript and abstract was just presented at ESC 18
Author Response
1) Can the author disclose that this work has been presented in the ESC 2018! Understand this is the manscript and abstract was just presented at ESC 18
We thank the reviewer for pointing out, that the data was presented at ESC 2018. We have added a sentence to the disclosure statement on page 8.
Reviewer 2 Report
In this manuscript,Kahr et al investigated an association between colorectal cancer(CRC) and atrial fibrillation(AF) in the setting of routine screening colonoscopy in 1949 asymptomatic participants. The results showed that a higher prevalence of advanced colorectal adenomas and carcinomas in patients with AF undergoing screening colonoscopy and suggests that cardiologists should especially consider CRC screening in AF patients.This manuscript is well written with rather good design. This is an interesting study, but there are some issues need to be addressed.
Major comments:
Should a diagnosis of AF prompt a search for occult cancer?
In this manuscript,author suggests that cardiologists should especially consider CRC screening in AF patients. Rahman F et al(JAMA Cardiol. 2016 July 1) and Vinter N et al(J Am Heart Assoc. 2018;7) do not support recommendations for using new-onset AF as an indication for systematic cancer screening . Author should discuss this issue in this manuscript.
Minor :
Page 1. Abstract : line 15 , 63.0% vs 37.1%-----> 63.0% vs 33.4%
Page 4. line 16: 29.4%-------> 28.4%
Page 4. line 25: the unmatched AF and non- AF groups---------->the AF and unmatched non-AF groups
Supplementary Table 1: page 2: Sigma-------> Sigmoid
Author Response
Reviewer 2:
1) Major comments: Should a diagnosis of AF prompt a search for occult cancer? In this manuscript,author suggests that cardiologists should especially consider CRC screening in AF patients. Rahman F et al (JAMA Cardiol. 2016 July 1) and Vinter N et al (J Am Heart Assoc. 2018;7) do not support recommendations for using new-onset AF as an indication for systematic cancer screening . Author should discuss this issue in this manuscript.
We thank the reviewer for raising this important aspect. Due to the design and limitations of the current study, our data does not support CRC screening outside of the patient spectrum, in which CRC screening is recommended anyway. Having said that, we believe that this study along with previously published reports, demonstrates a strong association between atrial fibrillation and CRC and other forms of cancer. Hence, general practitioners and cardiologists should screen their atrial fibrillation patients for their CRC screening status and, if they have not been adequately screened, refer them for screening. We have added this statement to the discussion section on page 8. Also, we have changed the final sentence of the abstract:
NEW: Our data suggests that practitioners should monitor the CRC screening status especially in AF patients.
2, 3, 4) Minor : Page 1. Abstract : line 15 , 63.0% vs 37.1% -----> 63.0% vs 33.4%; Page 4. line 16: 29.4%-------> 28.4%; Page 4. line 25: the unmatched AF and non- AF groups---------->the AF and unmatched non-AF groups
We thank the reviewer for identifying these typos. We have corrected them in the current version of the manuscript.
5) Supplementary Table 1: page 2: Sigma-------> Sigmoid
Changed.